# Extraction and Mass Spectrometric Characterization of Terpenes Recovered from Olive Leaves Using a New Adsorbent-Assisted Supercritical CO_2_ Process

**DOI:** 10.3390/foods10061301

**Published:** 2021-06-05

**Authors:** Zully J. Suárez Montenegro, Gerardo Álvarez-Rivera, Jose A. Mendiola, Elena Ibáñez, Alejandro Cifuentes

**Affiliations:** Foodomics Laboratory, Institute of Food Science Research (CIAL, CSIC), Nicolas Cabrera, 9, 28049 Madrid, Spain; zully.suarez.montenegro@gmail.com (Z.J.S.M.); gerardo.alvarezrivera@gmail.com (G.Á.-R.); j.mendiola@csic.es (J.A.M.); a.cifuentes@csic.es (A.C.)

**Keywords:** GC-QTOF-MS, supercritical CO_2_ extraction, terpenes, olive leaves, adsorbent-assisted processes, agricultural by-products

## Abstract

This work reports the use of GC-QTOF-MS to obtain a deep characterization of terpenoid compounds recovered from olive leaves, which is one of the largest by-products generated by the olive oil industry. This work includes an innovative supercritical CO_2_ fractionation process based on the online coupling of supercritical fluid extraction (SFE) and dynamic adsorption/desorption for the selective enrichment of terpenoids in the different olive leaves extracts. The selectivity of different commercial adsorbents such as silica gel, zeolite, and aluminum oxide was evaluated toward the different terpene families present in olive leaves. Operating at 30 MPa and 60 °C, an adsorbent-assisted fractionation was carried out every 20 min for a total time of 120 min. For the first time, GC-QTOF-MS allowed the identification of 40 terpenoids in olive leaves. The GC-QTOF-MS results indicate that silica gel is a suitable adsorbent to partially retain polyunsaturated C10 and C15 terpenes. In addition, aluminum oxide increases C20 recoveries, whereas crystalline zeolites favor C30 terpenes recoveries. The different healthy properties that have been described for terpenoids makes the current SFE-GC-QTOF-MS process especially interesting and suitable for their revalorization.

## 1. Introduction

Olive leaves (*Olea europaea L*.) are an important agricultural residue in Spain and other olive oil-producing countries; they come from both the mechanical pruning of the trees and the cleaning processes of the olive during the harvest of the fruit. This agricultural waste represents 25% of the total biomass generated in the olive oil industry [1], which accounts for over 500,000 t per year in Spain [2].

Several studies have reported important bioactive properties of compounds from olive leaves [3], thus being considered as a promising natural source not only for its functional value but also for cosmetic and pharmaceutical industries. The bioactivity of olive leaves has been traditionally associated to its content in phenolic derivatives [4,5], flavonoids, and terpenoids [6,7,8,9], among others. These compounds provide a wide range of health-related properties [10,11] such as anti-inflammatory [12,13] antioxidant [14], antitumor, and anticancer [15,16].

A broad variety of terpenoid compounds, including monoterpenes (C10), sesquiterpenes (C15), diterpenes (C20), and triterpenes (C30) have been described in olive leaves. Phenolic monoterpenes such as thymol have been reported as antioxidant, anti-inflammatory [17,18], and antimicrobial compounds [19,20]. Koc et al. [21] studied the gastroprotective effects of thymol and oleuropein, which are both bioactive compounds present in olive leaves. β-caryophyllene, within the sesquiterpenes family (C15), shows antioxidant, anti-inflammatory, anticancer, and neuroportective activities [22]. Aghajani et al. [3] identified the presence of this volatile compound in leaves of two olive varieties of olive leaves (mission and conservolea). Within the C20 terpene family, several studies reported the presence and bioactive potential of α-tocopherol in olive leaves [23], whose enrichment was achieved employing supercritical fluid extraction (SFE) [24,25,26]. With regard to the triterpenes family (C30), β-amyrin, α-amyrin, and uvaol have been reported as the main compounds in olive leaves [6,7,23] with several associated biological properties such as anti-inflammatory, antimicrobial, antifungical, antiviral, anticancer, and anti-ulcer [23,27,28,29,30].

It is interesting to mention that in almost all of the above described examples in the literature, the studies focused on the characterization of a reduced group of terpenes or even of a single molecule and, therefore, there is not a global approach to identify the expected large number of terpenes from olive leaves. The most recent work published in 2021 presented a metabolomics approach based on GC-MS combined with a simultaneous esterification–silylation reaction; however, the number of terpenes found in olive leaves was rather low (seven triterpenes and one monoterpene) [31]. In the current work, we used gas chromatography coupled to high-resolution mass spectrometry (GC-QTOF-MS) focusing on the identification of terpenes molecules.

In order to obtain bioactive compounds from olive leaves with high added value, alternative extraction technologies are required to substitute the conventional extraction procedures. Supercritical fluid extraction (SFE) is a green process with high energy efficiency, low toxicity, and appropriate physicochemical properties such as density, diffusivity, viscosity, and dielectric constant [32]. Although SFE has been widely employed for terpene extraction and fractionation [33], only a few reports can be found concerning SFE of terpenes from olive leaves; in fact, a search done in the Scopus database using (supercrit* AND extract* AND olive AND leave*) shows only three papers published on this subject [26,34,35]. Combined processes including supercritical extraction and adsorption have been also employed to selectively enrich different terpenoids’ fractions from a wide range of raw materials [36,37,38,39]; nevertheless, none of them use olive leaves as a biomass.

In this context, the aim of the present work was to carry out a deep characterization of terpenoid compounds from olive leaves that could provide a basis to differentiate and substantiate the related health benefits. To do this, in this work, GC-QTOF-MS is combined with supercritical fluid extraction and dynamic adsorption/desorption in order to obtain and characterize different fractions from olive leaves with a tailored composition of terpenes. For this purpose, the new process was designed using different adsorbents materials, which were selected based on their physicochemical properties, commercial availability, and low costs.

## 2. Materials and Methods

### 2.1. Vegetable Material

Shade-dried olive leaves (Cornicabra variety, harvested in January 2018, from Albacete (Spain)) with humidity lower than 10% were supplied by a local producer (Murciana de Herboristería S.A., Murcia, Spain). Branches and other impurities were removed manually from leaves before grounding with a knife mill at room temperature (Retsch Grindomix Ref GM200-Germany) at 8000 rpm for 40 s and sieved to 500–1000 µm particle size using an electromagnetic sieve shaker (CISA Sieving Technologies BA-200N, Barcelona, Spain).

### 2.2. Adsorbent Material

Different types of adsorbents as shown in Table 1 have been tested in this work for their suitability to selectively enrich targeted terpenes and/or families of terpenes. The pore size, particle size, and surface area of studied adsorbents are summarized in Table 1, as reported by the manufacturer. Moreover, the apparent density or bulk density (ρ_b_) was calculated for the target adsorbents according to the following formula ρ_b_ = w/V; where w is the mass of adsorbent placed in the column bed, and V is the volume of the column filled with the adsorbent. It is important to highlight that the amount of adsorbent employed to fill out the adsorption column (for a fixed bed length and column diameter) is different for each material and depends on the adsorbents’ characteristics. Thus, the use of real bulk density provides more valuable data on the mass of adsorbent present in the column and, therefore, on the number of adsorption sites within the column in each of the tested processes. Adsorbent materials used in this study were silica gel (Sigma-Aldrich, Sarajevo, Bosnia and Herzegovina), zeolite Y, ammonium (Alfa Aesar, Karlsruhe, Germany) and aluminum oxide 150 Tipe T (Merck, Germany). Sea sand (VWR Chemical BDH, Leuven, Belgium) was used mixed with zeolite (sea sand/zeolite, 2:1) to avoid adsorption column caking because of the high surface area of this material.

### 2.3. Supercritical Fluid Extraction

The extraction was carried out in a Speed Helix supercritical fluid extractor from Applied Separations (Allentown, PA, USA) using neat CO_2_ (Carburos Metálicos, Air Products Group, Madrid, Spain) as solvent. SFE starting conditions were based on previous studies performed on olive leaves [26,34]. In brief, 50 g of olive leaves and 100 g of sea sand were mixed and loaded inside a basket placed into the stainless-steel extraction cell; a filter paper was used to retain the material inside the basket. Extraction parameters were fixed at 30 MPa, 60 °C, and a constant flow rate of 9 L min^−1^ CO_2_ gas. Extraction kinetics was studied for 120 min taking a sample every 20 min. The extraction yield curve was constructed considering total yield (%) ($\frac{g\;of\;olive\;leaf\;extract}{g\;of\;dry\;material\;}\cdot \;100$) vs. extraction time.

### 2.4. Adsorption Process

A stainless-steel cylindrical adsorption cell (29 cm length and 0.65 cm i.d., for a total column volume of 38.5 cm^3^) was installed in the extraction/adsorption system, as shown in Figure 1. The extraction/adsorption/desorption process was carried out dynamically for 120 min at the fixed pressure and temperature conditions selected for extraction (30 MPa and 60 °C). The adsorbent material was packed into the second cell (adsorption column) with glass wool and high-quality cellulose disk filters at the entrance and exit of the cell to prevent plugging. Carbon dioxide passed through the supercritical extraction cell, and the solute extracted was adsorbed by the packed material in the adsorption column connected in series, as a dynamic mass transfer process under the same pressure and temperature conditions. Fractions were collected every 20 min at the exit of the adsorption column, after depressurization through an expansion valve (Parker Autoclave Engineers, Erie, PA, USA). After 120 min, complete depressurization took place for 30 min.

In order to recover the compounds not desorbed during the process and still remaining in the adsorbent, the material was washed with high-purity grade ethanol (VWR Chemicals-BDH, Fontenay-sous-Bois, France) by agitation at room temperature for 2 h, and extracts obtained were filtered. This treatment was applied for all materials except for zeolite; in this case, a further centrifugation step (Eppendorf centrifuge 5804R, Hamburg, Germany) at 10,000 rpm for 10 min was necessary. The supernatant passed through a filter of 0.45 µm pore size and 13 mm diameter. All experiments were done in duplicate. The extraction yield (%) of all fractions w expressed on a dry weight basis.

Recovery values (%) for a particular terpenes family ($C_{i}=C10,\;C15,\;C20,\;\mathrm{or}\;C30$) obtained at a defined extraction time (*t* = [0–20], [20–40], [40–60], [60–80], [80–100], or [100–120] min), using a certain type of adsorbent material (*s* = S60, S60P, S150, S150P, ZeAmG, or AO, see Table 1) were calculated as follows:(1)$$\%\;Recovery\;\left[C_{i\left(t,s\right)}\right]=\frac{A_{Ci\left(t,s\right)}}{\Sigma A_{C_{i}\left(control\right)}}*100$$
where $A_{Ci\left(t,s\right)}$ is the abundance of the target terpenes family extracted under fixed conditions of time and adsorbent; and $\Sigma A_{Ci\left(control\right)}$ is the sum of abundances of all terpene families (total terpenes abundance) obtained under control conditions (*t* = 120 min, without adsorbent).

### 2.5. GC-QTOF-MS Analysis

The analysis of the SFE extracts and fractions was performed employing an Agilent 7890B gas chromatography (GC) system coupled to an Agilent 7200 quadrupole time-of-flight (Q-TOF) analyzer (Agilent, Santa Clara, CA, USA) controlled using Mass Hunter software (Qualitative version 10.0 and Quantitave version 10.1), which was equipped with an electronic ionization (EI) interface. The separation was carried out using an Agilent Zorbax DB5- MS + 10 m Duragard Capillary Column (30 m × 250 μm i. D. × 0.25 μm). Helium was used as carrier gas at a constant flow rate of 0.8 mL min^−1^. The injection volume was 1 μL. Splitless mode was used for injection, keeping the injector temperature at 250 °C. The GC oven was programmed at 60 °C for 1 min; then, it increased at a rate of 10 °C/min to 325 °C and was held at this temperature for 10 min. An MS detector was operated in full-scan acquisition mode at an *m*/*z* scan range of 50–600 Da (5 spectra per second). The temperatures of the transfer line, the quadrupole, and the ion source were set at 290, 150, and 250 °C, respectively.

Target terpenes were tentatively identified by systematic mass spectra deconvolution and search in the MS database, using the Agilent Mass Hunter Unknown Analysis tool (Mass Hunter Unknown software version 10.2), and the NIST 20 Mass Spectral database was used for MS search. All samples were analyzed at the same concentration level (10 mg/mL) in ethanol. Quantitative results for target terpenes were expressed in terms of relative abundance per g of extract. Terpenoids such as thymol, squalene, phytol, alpha-tocopherol, alpha-amyrin, uvaol, and erytrhrodiol were confirmed with the reference standard.

### 2.6. Statistical Analysis

All experimental results are given as mean ± standard deviation. Data treatment and figures were made with Microsoft Excel 365 (Microsoft, Washington, DC, USA). Regarding the multivariate data analysis, a compound-abundance table, including samples in columns, was submitted to cluster analysis and heatmapping using freely available web server Heatmapper (www.heatmapper.ca, accessed on 4 June 2021). A data matrix was previously scaled using an auto-scaling approach; that is, the data were mean-centered and divided by the standard deviation of each variable. A hierarchical clustering was applied using a complete linkage clustering method with Pearson distance measurement.

## 3. Results and Discussion

### 3.1. ScCO_2_ Extraction of Total Terpenes

In order to obtain fractions enriched in different families of terpenes from a natural source, different approaches can be envisaged. Among them, a complete extraction at conditions able to solubilize the main targeted terpenes together with a simultaneous time-dependent fractionation is proposed in the present work as a first approach. Previous results reported by other authors working with olive leaves and targeting terpenes were first considered [26,33,34], and therefore, we carried out the preliminary experiments at low pressures (10–20 MPa) and constant temperature of 45 °C for 120 min. Similar composition in terms of families of terpenes (%) were obtained at the different conditions tested, while total yield increased by increasing the pressure. Considering these data and the previous studies, the selected working conditions were set at 30 MPa and 60 °C for a more complete extraction of total terpenes. These conditions were considered as control, showing an extraction yield of 0.70 ± 0.03%.

A kinetic study was next carried out for 120 min, taking samples every 20 min in order to have a deeper knowledge on the possibility of performing a time-dependent fractionation for enriching the extracts in specific terpenes. Thus, extracts recovered every 20 min were analyzed by GC-QTOF-MS, and a group of representative terpenes was selected in this initial stage for the optimization study. Namely, Table 2 shows the selected representative compounds of the different families of terpenes identified in olive leaves’ extracts. The compounds identified were classified according to their structural similarity as monoterpenes (C10), sesquiterpenes (C15), diterpenes (C20), and triterpenes (C30); this classification allows us an easier discussion of the results obtained below.

### 3.2. Time-Dependent scCO_2_ Fractionation of Terpenes from Olive Leaves

The analysis of the obtained fractions from the kinetic study at 20, 40, 60, 80, 100, and 120 min revealed a small differential contribution of each terpene family (%) to each one of the mentioned six fractions, suggesting a certain degree of selectivity during the time-dependent fractionation process, as shown in Figure 2.

Figure 2 shows the total extraction yield (%) and the terpene family abundance (%) per fraction of the control experiment. Fractions are expressed in terms of collection time (20 to 120 min) and solvent mass/feed mass d.b. (S/F ratio) (from 6.5 to 39). As can be seen, at the beginning of the process, the extraction yield shows a linear behavior up to 80 min (26.0 S/F ratio), corresponding to a constant extraction rate (CER) period limited by the solubility of the easily accessible solutes in scCO_2_ under these conditions. After this period, the extraction rate is slightly lower and driven by the diffusion of the solvent inside the particles and the diffusion of solutes and solvent to the surface; this is considered the diffusion-controlled rate period (DCR). In this particular extraction process, only these two periods have been identified. As for the global extraction of terpenes, it can be seen that some of them were placed in the surface of the particles and were readily available for extraction; their concentration decreased after 20 min extraction and continuously increased until the easily available material is exhausted (min 80 of the extraction process).

This is in agreement with the global extraction yield showing that terpenes were the main contributors in this first period. After 80 min, terpenes were more difficult to extract, but since extraction yield kept increasing, it is easy to infer that the other extractable material was co-extracted together with terpenes. Although a plateau was not reached at the end of the process, the smaller slope of the second period demonstrated that it was not worth increasing the solvent consumption for a very small improvement of both total yield and total terpenes. A similar situation was observed by De Lucas et al. [26] studying three pressure conditions (25, 35, and 45 MPa) and demonstrating that the highest tocopherol recovery was obtained at 25 MPa; the authors mentioned that an increase of pressure up to 35 and 45 MPa decreased tocopherol recovery due to the competitive extraction of tocopherol and other matrix compounds.

In order to analyze the possible time-dependent fractionation of terpenes obtained as a function of CO_2_ consumed in the process, the abundance (%) of each terpene family in each scCO_2_ fraction was evaluated (calculated as Abundance of each terpene family at the selected time/Sum of abundances of all terpene families after 120 min extraction ×100). Figure 2 provides information about the different availability of terpenes in the fractions. In this sense, although C30 terpenes are the major compounds in all fractions, at the beginning of the process (6.5 S/F), a relatively smaller proportion of C30 compounds were extracted, and lighter terpenes such as C10 were preferentially recovered (together with C15 and C20). The concentration of C30 terpenes increased with the extraction time, reaching a final contribution close to 73% of the total terpene content at 120 min or 39.0 S/F. This behavior is most likely related not only to the solubility of the different families of terpenes but also to their distribution within the structure of olive leaves.

Therefore, even if observing some time-dependent fractionation, the relative composition in terpenes of the different fractions is quite similar and, consequently, not significant differences in bioactivities could be expected. In this sense, as mentioned above, distinct biological activities have been associated to the presence of different families of terpenes [40,41,42,43,44,45,46,47,48,49,50,51,52,53,54,55,56], and therefore, a better separation of these terpenes is mandatory. Moreover, a different selectivity would make the identification of the different terpenes easier.

### 3.3. Adsorbent-Assisted scCO_2_ Fractionation of Olive Leaves

In order to achieve extracts enriched in a particular class of terpenes, a dynamic supercritical extraction–adsorption–desorption process was developed. Therefore, an on-line sequential process consisting of supercritical fluid extraction at 30 MPa and 60 °C and continuous adsorption in a column (filled with the adsorbents shown in Table 1) under supercritical conditions was studied to get a selective enrichment of bioactive terpenes. The process has been described in the experimental section as a dynamic process in which extraction is carried out under supercritical fixed conditions for 120 min. During this time, compounds extracted are sequentially (and selectively) retained in the adsorption column (see Figure 1), depending on the physicochemical characteristics of the extracted solutes and adsorbents.

As mentioned, adsorption occurs under the same supercritical conditions (30 MPa, 60 °C), since depressurization takes place after the adsorption column, in the micro-metering valve. It is important to highlight that the process, as it is designed, includes the simultaneous desorption of compounds depending on their retention in the column and on the amount of scCO_2_ circulating through the adsorption column. Therefore, adsorption and desorption are two competitive processes that take place together with the extraction of the compounds from olive leaves once the system is running. In this sense, the process can be described as follows:(1)Extraction process: in a first step, solutes dissolve in scCO_2_ according to its solubility and following the kinetics extraction process shown in Figure 2;(2)Adsorption process: at the same time, the dissolved compounds interact with the adsorbent through a partition process [38] that depends on the solubility of the compound(s) in scCO_2_, the chemical surface of the adsorbent, pore size, apparent density, and surface area, among other parameters. Therefore, if affinity of the solutes for the adsorbent is higher than the affinity for scCO_2_ (solubility), compounds will be retained in the column;(3)Desorption process: scCO_2_ removes solutes from the adsorbent; when the amount of CO_2_ increases, there is a displacement of the equilibrium toward CO_2_, and adsorbed solutes leave the column and are recovered in the different fractions along the 120 min. Later on, after the processing time, adsorption column is left for 20 min for complete depressurization and the adsorbent is maintained for 2 h in contact with ethanol (with stirring at room temperature) to obtain the last terpene fraction named adsorbate.

Table 3 compiles the results obtained for the different adsorbents in the dynamic extraction–adsorption–desorption supercritical process. It includes the total extraction yield (%) and the relative percentage of total terpenes recovered after 120 min of the process (a sum of the different fractions eluted vs. total abundance of terpenes extracted in the control) and the relative amount (%) of terpenes extracted from the adsorbent after 2 h of cleaning with ethanol (adsorbates). Moreover, the distribution (%) of the total terpenes in the different families (C10, C15, C20, and C30) is included for a better discussion of the results. By considering the global extraction yield achieved when no adsorption column was employed (that is 0.7%), the lower yields obtained with the different adsorbents are indicative of the adsorption capacity of the different materials, meaning that a higher yield of adsorbent 1 compared to adsorbent 2 implies a weaker retention of the compounds extracted in the first material. Moreover, the ability for a selective adsorption can be inferred from data on terpenes’ recovery, as shown in Table 3.

Silica gel has been proposed as one of the most suitable adsorbents for the separation of complex mixtures of non-oxygenated and oxygenated terpenes [57,58,59,60]. By analyzing the data of Table 3 corresponding to Silica gel 60 Å and 150 Å (S60, S60P, S150, and S150P; adsorbent’s characteristics shown in Table 1) compared to control in terms of total extract yield, we can deduce that a different degree of adsorption is provided by the adsorbents, ranging from 0.16% (S60) to 0.68% of total yield (for S60P). The behavior is consistent for S150 and S150P, thus showing a direct effect of particle diameter (or bulk density) on the capacity of retention of the materials.

As for the terpenes’ recovery, the behavior is different in terms of retention of the different classes of compounds in the adsorbent. As can be seen, S60 and S60P (with smaller pore size) are able to retain a large amount of terpenes, and therefore, recoveries are very low (16.2 and 15.4%, respectively); moreover, compounds seem to be irreversibly retained, since a very small recovery was obtained after washing with ethanol for 2 h (12.7 and 13.4%, respectively).

In order to better visualize this behavior, Figure 3 shows the comparison between S60 (A), S60P (B), S150 (C) S150P (D), AO (E), and ZeAmG (F) in terms of yield (%) and total terpenes recovery as a function of S/F ratio. Color bars have been included for the identification of the relative composition of each fraction in terms of C10, C15, C20, and C30 recovery. As can be seen, the kinetics of the process for S60 follows an S-shaped curve reaching around 23% of the total yield (i.e., 0.16%) achieved in the control (0.70%) that seems to be related to the recovery of terpenes. For S60P, the shape of the kinetics curve clearly shows an adsorption up to a value of S/F of 26 (80 min) and a complete desorption when breakthrough volume was reached, although it is not related with the terpenes extracted from the olive leaves, since a maximum recovery around 20% was achieved for these compounds. As for the type of compounds preferentially adsorbed, C10 terpenes, which are the more polar, are irreversibly retained at the beginning of the process, while C30 terpenes were retained mainly between 60 and 100 min. Concerning S150 and S150P, the results obtained are consistent with the smaller apparent density of S150P, being the lowest of all silica tested. Total yield was around 90% of the control kinetics, with the retention taking place after 80 min, although, as can be seen in terms of terpenes’ recovery, the adsorption was not related to this kind of compounds but to other solutes co-extracted from the olive leaves.

Figure 3 also shows the behavior of AO and ZeAmG. As for the total yield, AO behaves as a stronger adsorbent than zeolite and intermediate between S60 and S150, representing around 36% of the global yield of the total kinetics (control: 0.70%); this can be related to the highly porous and amorphous structure of AO.

Even if lower yields are obtained using AO, the recovery of terpenes is relatively high, and it is higher than that of S60, S60P, and S150, although it is lower than S150P and ZeAmG. Analysis of the composition of the fractions obtained after the whole process indicates that at the beginning of the process with AO, C10 polar compounds are retained preferentially, and that between 40 and 80 min, there is a competence between C20 and C30 for the active sites of the adsorbent. As for zeolites (ZeAmG), the terpenes’ elution profile is quite similar to the control kinetics, showing some degree of retention of the small molecular weight terpenes at the beginning of the process. In summary, Figure 3 shows that the adsorption/desorption process introduces an important selectivity to the SFE process, which should allow obtaining fractions with a tailored terpenes’ composition, as will be discussed in the next section.

### 3.4. Selective Enrichment of Terpenes in Adsorbent-Assisted scCO_2_ Extracts

A comparative view of the adsorption capacity of the studied adsorbents to selectively retain different types of terpenoids is shown in Figure 4. Although, as expected, the global terpenoids recovery values obtained in the proposed adsorbent-assisted extraction processes were lower compared to control conditions—no adsorption—(Figure 4A), a detailed view on the recovery values of different terpene families (normalized recoveries) show significant differences compared to control, allowing us to draw some conclusions about the selectivity and relative adsorption capacity exhibited by the adsorbents.

Due to the polar nature of target monoterpenes (C10) and sesquiterpenes (C15) (lowest log K_ow_), the enrichment degree for these compounds in the extracts obtained by adsorbent-assisted procedures are comparatively lower than under control conditions, as they are expected to be retained in the tested polar adsorbents (based on silica and alumina). However, some adsorbent materials exhibit certain affinity for these families of compounds. Thus, higher recoveries of C10 terpenes are obtained using ZeAmG and AO (lower retention) in comparison with S60, S60P, and S150 silica gel materials, which showed higher retention capacity. The stronger affinity shown by silica-based materials for polar compounds can be explained by the higher polarity of Si–O bonds compared to Al–O bonds. Interestingly, more enriched extracts in C15 terpenes are obtained with S150P than with ZeAmG and AO; whereas S60, S60P, and S150 also present a higher retention capacity of C15.

Unlike C10 and C15 terpenes, higher molecular weight terpenoids such as C20 and C30 undergo a significant enrichment in the extracts obtained with some selected adsorbents compared to extracts obtained under control conditions. As can be clearly seen in Figure 4B, adsorbents S60 and S60P, followed by AO and S150, increased the recovery values of C20 terpenes, whereas ZeAmG exhibited higher affinity for this family of compounds. On the other hand, C30-enriched extracts (around 80%) can be obtained using ZeAmG as adsorbent material, whereas silica gel-based adsorbents show a greater retention capacity of C30 terpenes (enrichment around 50%).

From the above-mentioned results, it seems clear that silicates are suitable adsorbent materials to retain polyunsaturated C10 and C15 terpenes and terpenols, removing these low molecular weight terpenes from the eluted extracts. Silicates also showed greater adsorption capacity of C10 and C15 terpenes compared to alumina (AO) and aluminosilicates (ZeAmG). The higher log K_ow_ of C20 and C30 terpenoids reduce the affinity of these higher molecular weight compounds for the polar adsorbent materials, leading to an enrichment of these compounds in the eluted extracts. Thus, amorphous and porous silicates (S60, S60P, S150) and alumina increase the recoveries of C20, whereas crystalline zeolites favors C30 terpenes recoveries.

### 3.5. GC-QTOF-MS Analysis of Terpenoid Compounds in Olive Leaves Extracts

SFE extracts and adsorbates were subjected to a comprehensive profiling analysis by GC-QTOF-MS (see Figure 5 for a total ion chromatogram of terpenoids identified in an olive leaves’ extract) to characterize the terpenoid composition. A total of 40 terpenes and terpenoids were tentatively identified on the basis of the positive match of the experimental mass spectra with MS databases (i.e., NIST and Fiehn lib), exact mass values as determined by HRMS, data reported in literature, and commercial standards when available. GC-QTOF-MS parameters such as retention time, generated molecular formula, match factor values from the MS database search, and main HRMS fragments are shown in Table 4. Annotated terpenoids were classified into families according to the number of isoprene units involved in the chemical structure; monoterpenoids (C10), sesquiterpenoids (C15), diterpenoids (C20), and triterpenoids (C30). Identification reliability was considered satisfactory for chemical structures, showing math factor values above 70.

Five phenolic monoterpenes and two bicyclic monoterpenoids were found among the most volatile compounds at early retention times (4.3–11.4 min). Two cymenol isomers (peaks 3 and 4, C_10_H_14_O) were the most abundant monoterpenoids; one of them was annotated as thymol (peak 4), with reported antioxidant, anti-inflammatory [17,18], and antimicrobial properties [19,20] in olive leaves extracts. Methoxylated (peaks 2 and 6) and dimethoxylated (peak 7) phenolic monoterpenoids were annotated as anethol, eugenol, and methyleugenol, respectively. According to the generated molecular formulae, the remaining two monoterpenes (peaks 1 and 5) exhibit an aliphatic and bicyclic structure, and these were annotated as borneol isomer (C_10_H_18_O) and camphene (C_10_H_16_), respectively.

A group of five terpenoids exhibiting C11 or C13 carbon atoms in their molecular formulae was identified in SFE extracts of olive leaves. These terpenoid derivatives, also known as apocarotenoids, are reported in the literature as natural degradation products of carotenoids, corresponding to the substituted cyclohexene ring moiety of the carotenoid’s framework, giving rise to a huge number of flavors and fragrances [61]. Peaks 12, 13, and 14 showed the same molecular formula (C_13_H_20_O_2_), with structural similarity to oxygenated derivatives of substituted cyclohexene ring (e.g., ionone and damascone), being tentatively identified as 4-oxo-β-isodamascol, 3-hydroxy-β-damascone, and cyclohexenone derivative, respectively. In addition, C11 terpenoids (peaks 10 and 17), showing mass spectra consistent with bicyclic lactones, were annotated as dihydroactinidiolide (C_11_H_16_O_2_) and isololiolide (C_11_H_16_O_3_), respectively. The formation of bicyclic derivatives through structural rearrangements of the substituted cyclohexene ring is reported as a common carotenoids’ degradation pathway [61].

Common sesquiterpenes (C_15_H_24_) such as farnesene (peak 8), γ-elemene (peak 15), and germacrene D (peak 32), as well as oxidated sesquiterpenoids, including nerolidol (C_15_H_26_O, peak 9), caryophyllene oxide (C_15_H_24_O, peak 11), and globulol (C_15_H_26_O, peak 16) have been identified in the studied olive leave extracts, which is in line with previous reports in literature [3,40]. Similar to monoterpenes, the levels of sesquiterpenoids represent a small portion of the total terpenoids abundance in SFE extracts. However, the levels of germacrene D in target extracts stand out from the other sesquiterpenoids.

Diterpenoids and diterpenoid derivatives, namely meroditerpenoids, represent the second most relevant group of terpenoids in olive leave extracts in terms of number of compounds identified and relative abundance. These phytochemicals were detected at retentions times longer than those of sesquiterpenoids and shorter than those of triterpenoids (16.4–26.9 min). The first eluted diterpenoids (peaks 19, 21, and 22) exhibit an aliphatic phytol-like fragmentation pattern and were annotated as hexahydrofarnesyl acetone (C_18_H_36_O), isophytol, and phytol (C_20_H_40_O), respectively, whereas peak 20 was assigned to geranylgeraniol (C_20_H_36_O), which is an aliphatic polyunsaturated structure. Hexahydrofarnesylacetone is a well-known degradation product of phytol, which is a diterpene alcohol that occurs as a side chain of chlorophyll a in all plants [3]. In addition, a subgroup of six terpenoid derivatives, showing a tocopheroid-like structure, was detected at later retention times (24.9–26.9 min). These compounds were classified as meroditerpenoids with a structure partially derived from terpenoid pathways, as they contain a phytyl moiety from chlorophyll degradation attached to a heterocyclic moiety from the shikimate pathway (homogentisate biosynthesis). Thus, peaks 24 and 25 were annotated as tocospiro A and B isomers (C_29_H_50_O_4_), containing a heterocyclic spiro moiety; whereas the MS fragmentation of peaks 26, 27, and 28 revealed the identity of γ, β, and α tocopherol (C_28_H_48_O_2_, C_29_H_50_O_2_), respectively, containing a substituted 2H-chromene moiety. Peak 29 was annotated as α-tocopherolquinone, which is a natural oxidation product of α-tocopherol. Furthermore, a substituted chromene derivative (peak 18, C_12_H_20_), most probably from the tocopherol degradation pathway, was tentatively identified. Meroterpenoids of the chromene class showed inhibitory activity toward butyrylcholine esterase [62].

The major group of compounds identified in olive leave extracts involves triterpenoids. Due to their lower volatility, these high molecular mass terpenoids have been mainly detected at the latest retention times (27.7–32.1 min). Squalene (peak 23, C_30_H_50_), the biochemical precursor of phytosterols and other non-steroidal triterpenes, was identified as one of the most abundant terpenes. Although at lower levels, two steroid-type structures (peaks 30 and 31) were found and annotated as stigmasterol and β-sitosterol, respectively. Pentacyclic triterpenoids with amyrenyl skeletons (oleanane and ursane types) were the major ones. Thus, five triterpenoids exhibiting mass spectra consistent with ursane-type structure were annotated as α-amyrin (peak 34, C_30_H_50_O), ursolic acid derivatives (peak 37 and 38, C_30_H_48_O_2_), erythrodiol, and uvaol (peak 39 and 40, C_30_H_50_O_2_). Two compounds from the oleanane group, namely β-amyrin (peak 34, C_30_H_50_O) and the acetylated derivative (peak 35, C_32_H_52_O_2_), as well the lupane-type triterpene derivative lupenol acetate (peak 36, C_30_H_48_O_2_) were annotated as highly abundant compounds in SFE on olive leaves. Figure 6 shows some representative chemical structures of terpenoid compounds from olive leaves extracts found in this work.

### 3.6. Differential Terpenoids Composition in Olive Leaves Extracts

The SFE extracts and adsorbates obtained in this work were comparatively evaluated in terms of terpenoids enrichment.

All extracts and adsorbates were grouped according to their relative terpenoids content after applying a clustering method to both rows and columns of the data matrix. The differential enrichment of terpenoids at different SFE conditions is depicted in the resulting heatmap displayed in Figure 7, which shows a color code from lower (light red) to higher concentration levels (light green). Although most terpenoids were detected in all samples, significant changes in the abundance of target terpenoids in the studied extracts were observed. According to the column dendrogram, samples can be classified into four different groups according to their terpenoids composition; i.e., two groups on the left side including silica, alumina, and zeolite extracts, the control group in the middle of the dendrogram, and a heterogeneous groups of samples on the right side of the graph, mainly containing the adsorbates. It is worth noting that extracts of silica (e.g., S60 and S150P), aluminum oxide (AO), and zeolite (e.g., ZeAmG20 and ZeAmG60) are classified in opposite sides to their corresponding adsorbates, which indicates the different composition of extracts and adsorbates.

The big cluster of silica, alumina, and zeolite extracts on the left side of the dendrogram is mainly characterized by lower levels of pentacyclic triterpenes (e.g., erythrodiol, uvaol, and ursolic acid derivatives) and apocarotenoids (e.g., 4-oxo-β-isodamascol, 3-hydroxy-β-damascone, cyclohexenone derivative, and isololiolide). However, remarkable differences are observed between subgroups of samples. Thus, the levels of pentacyclic triterpenes and apocarotenoids are moderately higher in ZeAmG extracts than in silica and alumina extracts, although in general, the levels of terpenoids are comparable to control extracts. The clusters of samples AO and S150P reveal the capacity of alumina and higher particle size silica gel to yield extracts enriched in mono-, sesqui-, and (mero)diterpenoids (i.e., tocopherols and other phytol derivatives). In contrast, lower levels of these low molecular weight terpenoids are obtained in S60 extracts, which is in line with the higher retention capacity of small particle size silica gel adsorbents, as discussed in Section 3.4. Interestingly, S150P extracts at 60–120 min show higher enrichment in phytosterols and amyrins compared to silica and alumina extracts.

Unlike the extracts, adsorbates are clustered on the right side of the dendrogram, showing lower levels of mono-, sesqui-, and (mero)diterpenoid, whereas higher enrichment levels of triterpenoids and apocarotenoids are obtained in general, although with some discrepancies. Thus, adsorbates are clustered in two subgroups. On the one hand, silica and alumina absorbates exhibit a significant enrichment in triterpenoids and apocarotenoids, whereas alumina and S150P silica gel show again a similar behavior with lower values of phytosterols and amyrins. On the other hand, ZeAmG adsorbate is classified in another subcluster, as the levels of retained terpenoids are notably lower in this sample. The lower enrichment of extracts obtained with a crystalline adsorbent such as zeolite is evidenced in the position of this sample in the dendrogram, which is classified as at a close distance to control samples. However, despite this behavior, zeolite exhibits a moderate to high selectivity toward pentacyclic triterpenoids such as erythrodiol, uvaol, and ursolic acid derivatives, as well as the diterpenoid phytol.

## 4. Conclusions

This work provides for the first time a deep characterization of the terpenoids fraction that can be extracted from olive leaves, which is one of the largest by-products generated by the olive oil industry. GC-QTOF-MS has been combined with a selective fractionation process based on dynamic online coupling of SFE and an adsorption/desorption process for extracting and identifying terpenoids from olive leaves (*Olea europaea* L.). For the first time, 40 terpenes and terpenoids have been identified.

Several commercial low-cost adsorbents with diverse chemical nature and physicochemical properties have been studied. The silica gel group showed a higher retention capacity of the C10-C15 terpene family. Aluminum oxide maximized the recovery of diterpenes (C20). Finally, triterpenes (C30) were recovered mainly using zeolite Y-ammonium. The fractionation and identification process is shown to provide extracts with different terpenoids composition, and therefore, diverse biological activities can be expected from them. Future work evaluating the neuroprotective activity of the different olive leaves’ extracts is being carried out in our laboratory.

## Figures and Tables

**Figure 1 foods-10-01301-f001:**
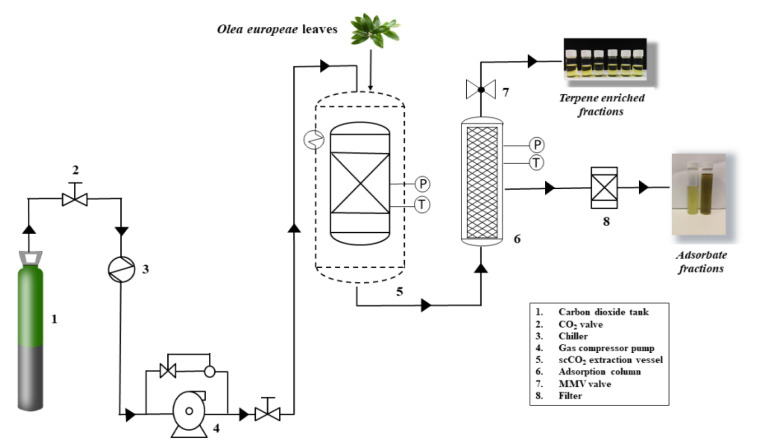
Scheme of the adsorbent-assisted supercritical CO_2_ extraction system.

**Figure 2 foods-10-01301-f002:**
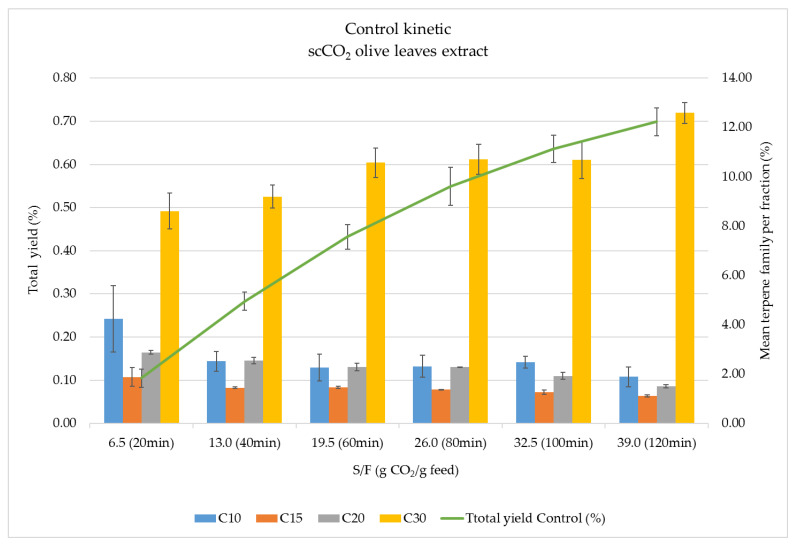
Total extraction yield (%) and terpene family abundance (%) in scCO_2_ extracts from olive leaves obtained at different fractionation times for the control experiment.

**Figure 3 foods-10-01301-f003:**
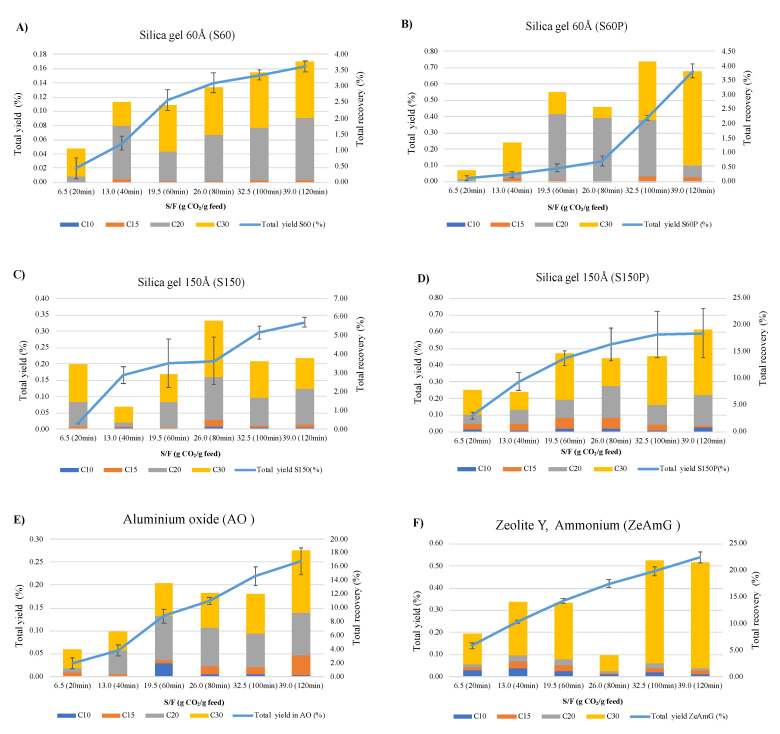
Comparison of total yield (%) and total terpenes recovery (%) in the studied adsorbents (**A**) S60; (**B**) S60P; (**C**) S150; (**D**) S150P; (**E**) AO; and (**F**) ZeAmG vs. S/F ratio.

**Figure 4 foods-10-01301-f004:**
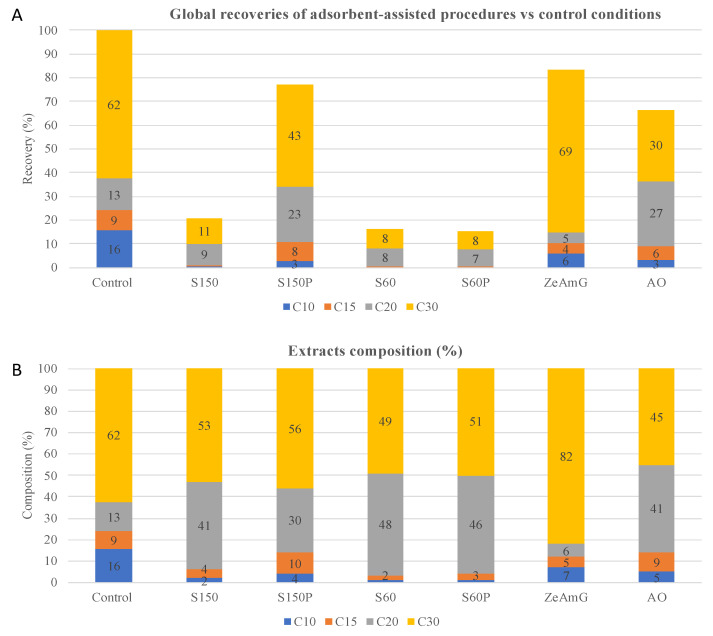
Global terpenoid families’ recoveries (**A**) and terpenoid families’ composition (**B**) of olive leaves extracts obtained by the studied adsorbent-assisted processes and under control conditions (no adsorption).

**Figure 5 foods-10-01301-f005:**
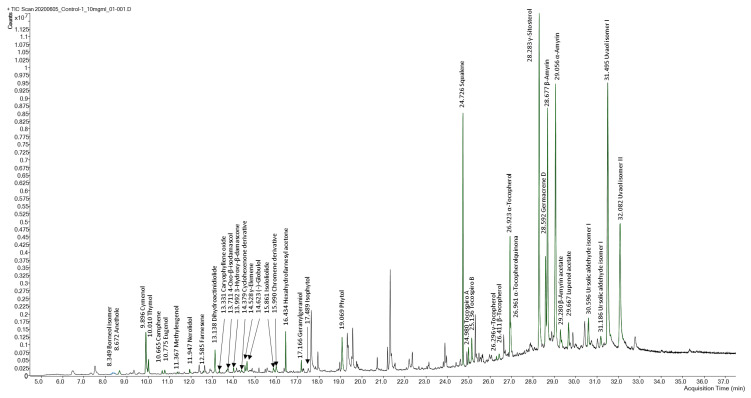
GC-q-TOF-MS total ion chromatogram of olive leave SFE extract obtained under control conditions (10 mg/mL). See peak assignment in Table 4. Non-terpenoid compounds identified: 1. n-Hexadecanoic acid; 2. Hexadecanoic acid; 3. cis,cis-7,10,-Hexadecadienal; 4. Octadecanoic acid; 5. 9-Octadecenamide, (z)-; 6. Cyclopropanebutanoic acid; 7. 9,12-Octadecadienoyl chloride, (z,z)-; 8. 6,9,12,15-Docosatetraenoic acid, methyl ester; 9. Myo-inositol.

**Figure 6 foods-10-01301-f006:**
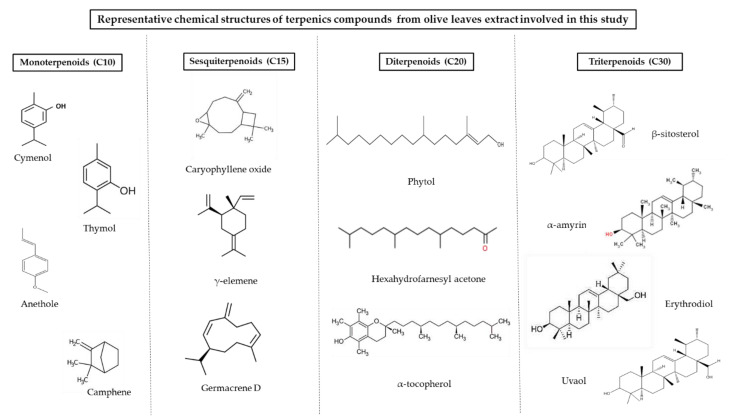
Representative chemical structures of terpenoid compounds from olive leaves extracts involved in this study.

**Figure 7 foods-10-01301-f007:**
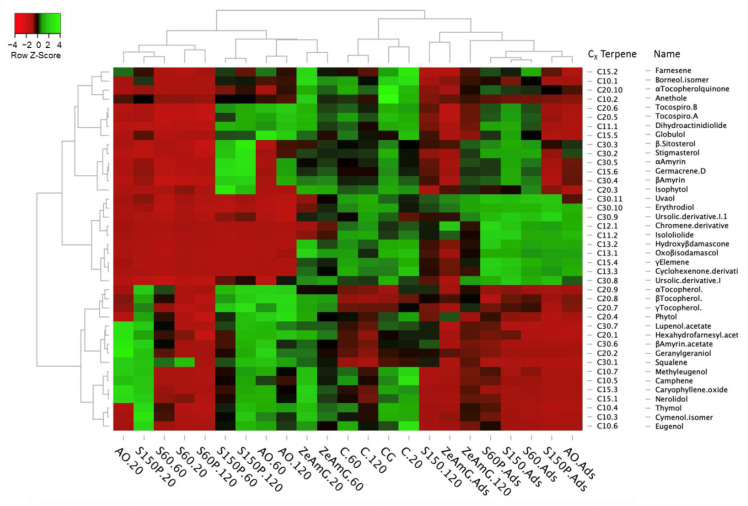
Heatmap showing the differential enrichment of identified terpenoid compounds (P1–P40) in olive leaves extracts and adsorbates obtained by SFE. Color code: light green (higher enrichment); light red (lower enrichment).

**Table 1 foods-10-01301-t001:** Summary of the most relevant adsorbent’s characteristics.

Name	Pore Size (Å)	Particle Size (mesh)	Surface Area (m^2^/g)	Bulk Density (mg/cm^3^)
Silica gel (S150)	150	35–60 (250–500 µm)	300	475.4
Silica gel (S150P)	150	200–425 (35–70 µm)	300	413.1
Silica gel (S60)	60	35–60 (250–500 µm)	480	810.6
Silica gel (S60P)	60	230–400 (40–63 µm)	530	613.1
Zeolite Y-ammonium * (ZeAmG)	n.r.	<125 µm	925	623.5
Aluminum oxide 150 Type T (AO)	58	70–230 (60–200 µm)	205	130.4

n.r. not reported; * Mole ratio: Ze:AmG = 5.1:1.

**Table 2 foods-10-01301-t002:** Representative terpenoids identified by GC-QTOF-MS from olive leaves extract obtained by scCO_2_, showing their molecular formula, molecular weight, number of rings, main functional groups in the molecule, Hansen Solubility Parameters (HSPs: contributions for dispersion (δD), polar (δP), hydrogen bond (δH), and total (δT)) and octanol–water partition coefficient (log(K_ow_)).

Family	Compound Name	RT (min)	Formula	MW (g/mol)	Number of Rings	Functional Group	δD	δP	δH	δT	log(K_ow_)
Monoterpenoid	Cymenol	9.89	C_10_H_14_O	150.22	1	-OH	18	4.9	9.1	20.8	2.5
Monoterpenoid	Thymol	10.01	C_10_H_14_O	150.22	1	-OH	17.8	4	7.2	19.6	3.37
Sesquiterpenoid	Germacrene	28.59	C_15_H_24_	204.35	2		17	1.6	2.6	17.3	6.44
Diterpenoid	Hexahydrofarnesyl acetone	16.43	C_18_H_36_O	268.5	0	-CO	16	3.2	2	16.4	7.02
Ditepenoid	α-Tocopherol (Vit E)	26.92	C_29_H_50_O_2_	430.7	1, 1-O	Phenol	16.9	1.5	3.6	17.3	11.06
Ditepenoid	Tocospiro A	25.13	C_29_H_50_O_4_	462.7	1, 1-O	-OH, -CO	16.5	5.1	3.8	17.7	7.39
Triterpenoid	Uvaol	31.49	C_30_H_50_O_2_	442.7	5	-CH_2_OH, -OH,	17.9	2.7	5.6	19	9.22
Triterpenoid	β-Amyrin	28.67	C_30_H_50_O	426.7	5	-OH	17.7	1.7	2.9	18.1	11.04

**Table 3 foods-10-01301-t003:** Total yield of the SFE process used as control vs. SFE using different adsorbents (the latter includes terpenes recovery in the fractions and in the adsorbent after completing the process).

Adsorbent	Total Extract Yield (%)	Terpenes Recovery Fractions (%)	C10 (%)	C15 (%)	C20 (%)	C30 (%)	Terpenes Recovery in Adsorbent
Control	0.70 ± 0.03	-	15.7	8.52	13.41	62.36	-
S150	0.33 ± 0.02	20.9	0.44	0.81	8.64	11.01	12.5
S150P	0.59 ± 0.14	77.1	2.83	8.06	23.17	43.05	30.9
S60	0.16 ± 0.01	16.2	0.04	0.36	7.78	8.00	12.7
S60P	0.68 ± 0.04	15.4	0.05	0.51	7.05	7.8	13.4
ZeAmG	0.54 ± 0.02	83.5	5.88	4.34	4.69	68.59	6.1
AO	0.25 ± 0.03	66.6	3.25	6.04	27.24	30.07	21.9

**Table 4 foods-10-01301-t004:** Terpenes and terpenoids in olive leaves identified by GC-QTOF-MS using NIST 20 Mass Spectral database.

Peak No	RT (min)	Family	Key	Tentative Identification	Formula	Match Factor	Main Fragments (*m*/*z*) ^b^	Reference
1	8.35	Monoterpenoid	C10#1	Borneol isomer	C_10_H_18_O	75	121, 110, 95	
2	8.67	Monoterpenoid	C10#2	Anethole	C_10_H_12_O	95	148, 133, 177, 105	
3	9.90	Monoterpenoid	C10#3	Cymenol isomer	C_10_H_14_O	80	135, 115, 91	[40]
4	10.01	Monoterpenoid	C10#4	Thymol ^a^	C_10_H_14_O	93	150, 135, 91	[63]
5	10.67	Monoterpene	C10#5	Camphene	C_10_H_16_	73	136, 121, 91	
6	10.78	Monoterpenoid	C10#6	Eugenol	C_10_H_12_O_2_	96	164, 149, 131, 103	[64]
7	11.37	Monoterpenoid	C10#7	Methyleugenol	C_11_H_14_O_2_	71	161, 119, 105	
8	11.95	Sesquiterpenoid	C15#1	Nerolidol	C_15_H_26_O	72	161, 133, 119, 105, 91	[64]
9	12.59	Sesquiterpene	C15#2	Farnesene	C_15_H_24_	68	133, 120, 93, 69	
10	13.14	Apocarotenoid	C11#1	Dihydroactinidiolide	C_11_H_16_O_2_	80	152, 137, 111	
11	13.33	Sesquiterpenoid	C15#3	Caryophyllene oxide	C_15_H_24_O	77	161, 136, 121, 107, 93	[3,40,64,65]
12	13.71	Apocarotenoid	C13#1	4-Oxo-β-isodamascol	C_13_H_20_O_2_	75	121, 105, 91, 93, 79	[3]
13	13.99	Apocarotenoid	C13#2	3-Hydroxy-β-damascone	C_13_H_20_O_2_	79	208, 193, 175	[3,64]
14	14.38	Apocarotenoid	C13#3	Cyclohexenone derivative	C_13_H_20_O_2_	73	161, 136, 121, 105	
15	14.53	Sesquiterpene	C15#4	γ-Elemene	C_15_H_24_	75	201, 132, 119, 83	[65]
16	14.62	Sesquiterpenoid	C15#5	(-)-Globulol	C_15_H_26_O	81	204, 189, 135, 109	
17	15.86	Apocarotenoid	C11#2	Isololiolide	C_11_H_16_O_3_	76	195, 152, 121	
18	15.99	Meroterpenoid	C12#1	Chromene derivative	C_12_H_20_	71	212, 197, 155	
19	16.43	Diterpenoid	C20#1	Hexahydrofarnesyl acetone	C_18_H_36_O	79	124, 109, 95	[3,66]
20	17.17	Diterpenoid	C20#2	Geranylgeraniol	C_20_H_34_O	80	135, 121, 107, 81	[3]
21	17.49	Diterpenoid	C20#3	Isophytol	C_20_H_40_O	85	123, 95, 81, 71	[3]
22	19.07	Diterpenoid	C20#4	Phytol ^a^	C_20_H_40_O	86	123, 95, 81, 71	
[3,23,25,35] 23	24.73	Triterpene	C30#1	Squalene ^a^	C_30_H_50_	90	410, 341, 136, 121, 109, 81	[23,25,35]
24	24.98	Meroditerpenoid	C20#5	Tocospiro A	C_29_H_50_O_4_	80	419, 402, 137	
25	25.14	Meroditerpenoid	C20#6	Tocospiro B	C_29_H_50_O_4_	79	419, 402, 137	
26	26.30	Meroditerpenoid	C20#7	γ-Tocopherol	C_28_H_48_O_2_	75	416, 191, 151	[23,26,67]
27	26.41	Meroditerpenoid	C20#8	β-Tocopherol	C_28_H_48_O_2_	76	416, 191, 151	[23,26,67]
28	26.92	Meroditerpenoid	C20#9	α-Tocopherol ^a^	C_29_H_50_O_2_	94	430, 205, 165	[23,26,35,67,68,69]
29	26.96	Meroditerpenoid	C20#10	α-Tocopherolquinone	C_29_H_50_O_3_	72	221, 203, 178, 150	
30	27.69	Triterpenoid	C30#2	Stigmasterol	C_28_H_48_O	73	400, 382, 315, 213	[70]
31	28.28	Triterpenoid	C30#3	β-Sitosterol	C_29_H_50_O	86	414, 329, 255, 213	[23,66,70]
32	28.59	Sesquiterpene	C15#6	Germacrene D	C_15_H_24_	75	204, 189, 175	
33	28.68	Triterpenoid	C30#4	β-Amyrin	C_30_H_50_O	88	218, 203, 189, 119	[7,23,67]
34	29.06	Triterpenoid	C30#5	α-Amyrin ^a^	C_30_H_50_O	81	218, 203, 189, 119	[7,23,67]
35	29.28	Triterpenoid	C30#6	β-Amyrin acetate	C_32_H_52_O_2_	90	218, 203, 189, 119	
36	29.67	Triterpenoid	C30#7	Lupenol acetate	C_30_H_48_O_2_	71	189, 161, 135, 121	
37	30.60	Triterpenoid	C30#8	Ursolic acid derivative I	C_30_H_48_O_2_	70	440, 273, 232, 135	[6,7,8,23,71]
38	31.19	Triterpenoid	C30#9	Ursolic acid derivative II	C_30_H_48_O_2_	75	203, 189, 175	[[6,7,8,23,71]
39	31.50	Triterpenoid	C30#10	Erythrodiol ^a^	C_30_H_50_O_2_	87	234, 203, 119	[7,30,72,73]
40	32.08	Triterpenoid	C30#11	Uvaol ^a^	C_30_H_50_O_2_	88	234, 203, 119	[6,7,8,23,30,66,71,73,74]

^a^ Identification confirmed by reference standard. ^b^ Quantitative *m*/*z* ion is underlined.

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
