# Peer review of "Extraction and Mass Spectrometric Characterization of Terpenes Recovered from Olive Leaves Using a New Adsorbent-Assisted Supercritical CO2 Process"

_foods, 2021, doi:10.3390/foods10061301_

Round 1

Reviewer 1 Report

The manuscript provides a comprehensive characterization of the terpenes fraction from olive leaves. The experimental section is well detailed, and results are clearly presented.

The manuscript is interesting from an analytical point of view, but novel insight from a biological perspective is not provided. For example, could the authors propose an innovative use of this by-products thanks to the bioactives they identified?

Introduction:

-Line 77: “In the current work, we will use” I would suggest using a past instead of a future sentence

Methods:

-I would add dome details on the sampling plan, such as geographical origin, variety and harvesting year. Indeed, secondary metabolites are strongly influenced by the environmental conditions, thus the result reported should be referred to a particular combination of cultivar, year, area. This result cannot be considered representative of the field variability in terms of secondary metabolism.

Results:

-As mentioned by the authors, a broad variety of terpenoid compounds have been already described in olive leaves. Would it be more interesting to enlarge the coverage of secondary metabolites to reach a comprehensive characterization of the olive leaves.

Author Response

The manuscript provides a comprehensive characterization of the terpenes fraction from olive leaves. The experimental section is well detailed, and results are clearly presented.

The manuscript is interesting from an analytical point of view, but novel insight from a biological perspective is not provided. For example, could the authors propose an innovative use of this by-products thanks to the bioactives they identified?

Authors want to thank reviewer 1 for considering that the results are clearly presented and that the experimental section is detailed. Moreover, thanks for considering that the work is interesting from an analytical point of view, which was our main objective in the present manuscript. As the reviewer mentions, considering terpenoids as target compounds identified in the manuscript, we have worked also in the biological activities, in this case related to the neuroprotective effect of the extracts enriched in target terpenes; this information is reflected in a second manuscript that has been sent for publication into Foods as a second part of the present manuscript (Manuscript ID: foods-1247012; Title: Neuroprotective Effect of Terpenoids Recovered from Olive Oil By-Products). Considering the huge amount of information given, we decided to divide the manuscript in two parts.

Introduction:

-Line 77: “In the current work, we will use” I would suggest using a past instead of a future sentence

We have modified the text according to the suggestion.

Methods:

-I would add dome details on the sampling plan, such as geographical origin, variety and harvesting year. Indeed, secondary metabolites are strongly influenced by the environmental conditions, thus the result reported should be referred to a particular combination of cultivar, year, area. This result cannot be considered representative of the field variability in terms of secondary metabolism.

Olive leaves were purchased to a local distributor of spices and herbs for food uses and tea preparation (Murciana de Herboristeria). According to their information, the leaves used in our study belong to Cornicabra variety, harvested in January 2018 in Albacete (Spain) during olive harvesting for oil production. Leaves were separated mechanically and naturally dried in shadow. This information has been included in the manuscript.

Results:

-As mentioned by the authors, a broad variety of terpenoid compounds have been already described in olive leaves. Would it be more interesting to enlarge the coverage of secondary metabolites to reach a comprehensive characterization of the olive leaves.

Indeed, we could have detected other volatile metabolites by GC-QTOF-MS or other more polar secondary metabolites by HPLC-QTOF-MS, however, the main focus of our research project is to investigate the terpenoids fraction of different food by-products, plants, algae and microalgae, to develop specific green extraction conditions for this type of compounds, to characterize them chemically and to investigate their bioactivity as a multitarget neuroprotective treatment of Alzheimer disease. To do this, we first accept that there is not cure for this pandemic yet and second, that the bioactivity of terpenoid compounds is different depending on their size and nature (e.g., antioxidant, anti-inflammatory, anti-cholinergic, etc), so, a combination of these compounds can provide the desirable multitarget approach, once their pass through the blood brain barrier has been demonstrated. So, as we have already mentioned, this work is just the beginning of a long project whose methodology, by the way, is everyday more followed by researchers of Alzheimer.

On the other hand, and also following the suggestions from Reviewer 2, we have included the information related to the identification of other intense peaks in Figure 5. In this sense, we have modified Figure 5 with the identification by GC-QTOF-MS of the rest of the peaks and include this additional information in the legend.

Reviewer 2 Report

The presented manuscript reports on the possibility of improved extraction and comprehensive analysis of terpenes recovered from olive leaves. Therefore, the subject addresses a praxis driven objective. The abstract is well prepared arousing an interest to the results presented in the manuscript. The chosen key words correspondingly reflect the research presented and the main outcomes of the study. The introductory state of art has been kept compact, allowing the reader to follow the derived objectives and gives a well-researched scientific background with reference to the biological effects of selected terpenes. Perhaps, in this context a few sentences to the general use of olive leaves e.g. in bread making could be elaborated, while going into possible limits with regard to taste and eventually known negative incidences.  The experimental design is clearly defined and well executed. The methods used are adequate, they represent standard methods generally applied and are technically sound and well described. The results are presented compactly/concisely; the discussion corresponds to the data presented. A total of 40 terpenes and terpenoids were tentatively identified on the basis of the positive match of the experimental mass spectra with MS databases. To this regard the authors could have enriched the manuscript by validating a few selected candidates by targeted analysis, while give some values of absolute quantification. In my opinion, the authors have still given altogether a very good report confining to essential findings indicating also many potentials for further work. The subject is interesting, praxis relevant, commercially attractive and will attract a good reader community.

Specific remarks

Line 47: … wide range of health related properties ...

Line 57: … shows ….

Line 59: … the presence of this volatile compound in leaves of two olive varieties of olive leaves (mission and conservolea).

Line 63: With regard to triterpene family (C30), …..

Line 70: … in literature, the studies focused ….

Line 86: … terpene extraction….

Line 96: … from olive leaves that could provide a basis to differentiate and substantiate the related health benefits.

Line 107-113: Was grinding done at room or at lower temperature? Please add accordingly to the method.

Line 117: … this work for their suitability …

Table 1: Second column – particle size – two values are given, what do they represent? Pore size?

Line 255: … and 60°C….

Line 256: I do not understand this sentence about the extraction yield:- “These conditions were considered as control, showing an extraction yield of 0.70±0.03%.” Why is it so low - or is it the amount extracted while being related to amount/mass of the leaves applied? In that case, please supplement the result once, so that the reader can better follow up.

Table 2: The Table contains values for d D, d P etc. I suppose these are Hansen Solubility Parameters (HSPs: contributions for dispersion, polar and H bond) and log (Kow). I would recommend to add these explanation in the Table legend, such that there is no need to refer it in the manuscript text. What is d T?

Figure 2: Again the terpene family abundance appears to be low with 12% or may be shortly explain on what basis the calculation was made ( - as explained in line 296?).

Lines 306-309: This part is slightly confusing being not clear and it perhaps needs to be re-written esp. with the part “… and steadily increased…..”

Line 335: …close to 73% ? Which values in the figure 2 are being referred to here?

Line 352: … consisting of supercritical fluid extraction ….

Line 414: What is “P” for?

Line 546: … showing match factor values…?

Line 724: … one of the largest …

Author Response

The presented manuscript reports on the possibility of improved extraction and comprehensive analysis of terpenes recovered from olive leaves. Therefore, the subject addresses a praxis driven objective. The abstract is well prepared arousing an interest to the results presented in the manuscript. The chosen key words correspondingly reflect the research presented and the main outcomes of the study. The introductory state of art has been kept compact, allowing the reader to follow the derived objectives and gives a well-researched scientific background with reference to the biological effects of selected terpenes. Perhaps, in this context a few sentences to the general use of olive leaves e.g. in bread making could be elaborated, while going into possible limits with regard to taste and eventually known negative incidences. The experimental design is clearly defined and well executed. The methods used are adequate, they represent standard methods generally applied and are technically sound and well described. The results are presented compactly/concisely; the discussion corresponds to the data presented. A total of 40 terpenes and terpenoids were tentatively identified on the basis of the positive match of the experimental mass spectra with MS databases. To this regard the authors could have enriched the manuscript by validating a few selected candidates by targeted analysis, while give some values of absolute quantification. In my opinion, the authors have still given altogether a very good report confining to essential findings indicating also many potentials for further work. The subject is interesting, praxis relevant, commercially attractive and will attract a good reader community.

We want to thank the reviewer for his/her nice words about our manuscript. As for the comment of validating a few selected candidates, this information can be found in the original manuscript (lines 230-232) and in Table 4, where we marked with a superscript those compounds which identification was confirmed by using a reference standard. 

Specific remarks

We want to thank reviewer for his/her remarks that have been appropriately addressed in the new version of the manuscript.

Line 47: … wide range of health related properties ...

Done.

Line 57: … shows ….

Done.

Line 59: … the presence of this volatile compound in leaves of two olive varieties of olive leaves (mission and conservolea).

Done.

Line 63: With regard to triterpene family (C30), …..

Done.

Line 70: … in literature, the studies focused ….

Done.

Line 86: … terpene extraction….

Done.

Line 96: … from olive leaves that could provide a basis to differentiate and substantiate the related health benefits.

Done.

Line 107-113: Was grinding done at room or at lower temperature? Please add accordingly to the method.

Done.

Line 117: … this work for their suitability …

Done.

Table 1: Second column – particle size – two values are given, what do they represent? Pore size?

Particle size is shown using two different units, mesh and µm.

Line 255: … and 60°C….

Done.

Line 256: I do not understand this sentence about the extraction yield:- “These conditions were considered as control, showing an extraction yield of 0.70±0.03%.” Why is it so low - or is it the amount extracted while being related to amount/mass of the leaves applied? In that case, please supplement the result once, so that the reader can better follow up.

Usually extraction yields in SFE are quite low because of its selectivity that is tuned by changing pressure and temperature providing with extracts enriched in the target compounds (and compounds with similar structure), while avoiding the extraction of other major compounds (that could contribute to achieve a higher yield).

Table 2: The Table contains values for d D, d P etc. I suppose these are Hansen Solubility Parameters (HSPs: contributions for dispersion, polar and H bond) and log (Kow). I would recommend to add these explanation in the Table legend, such that there is no need to refer it in the manuscript text. What is d T?

Table legend has been modified as suggested.

Figure 2: Again the terpene family abundance appears to be low with 12% or may be shortly explain on what basis the calculation was made ( - as explained in line 296?).

An explanation has been added in the text. In fact, values were obtained using Equation (1) but, in this case, they cannot be called %Recovery since no adsorbent was employed.

Lines 306-309: This part is slightly confusing being not clear and it perhaps needs to be re-written esp. with the part “… and steadily increased…..”

We are mentioning that the content “continuously” increased after 20 min and up to 80 min.

Line 335: …close to 73% ? Which values in the figure 2 are being referred to here?

We want to apologize since this number is not directly obtained from Figure 2. In fact, to get this number we consider the % of contribution of C30 at 120 min (abundance of C30) divided by the total terpene abundance at that time. It can be also seen as a relative % of contribution of C30 at 120 min.

Line 352: … consisting of supercritical fluid extraction ….

Done.

Line 414: What is “P” for?

“P” in S150 or S60P corresponds to a type of silica gel (with smaller particle diameter), different from S150 and S60 (this can be confirmed in Table 1).

Line 546: … showing match factor values…?

Our apologies but what is written is the correct form of expressing it.

Line 724: … one of the largest …

Done.

Reviewer 3 Report

The paper “Mass Spectrometric Characterization of Terpenes Recovered from Olive Leaves Using Adsorbent-Assisted Supercritical CO2 Fractionation” reports an experimental study focused to the extraction with SFE (Adsorbent –assisted) CO2  followed by a CC-QTof MS analysis with characterization based on Mass Spectra Databases.

In my opinion the paper can be accepted for publication after minor revision.

TITLE: In my opinion, the authors have to revise the title. By the present title seems that Mass Spectrometry characterization is central in the experimental work, but it is not (technical development of Adsorbent-Assisted Supercritical CO2 extraction is the central goal of the paper). Fractionation has to be changed in extraction, because it seems authors have fractionated the extract by density stepping or other procedures, while the object of the GC-MS analysis is a single extract, as showed in figure 5.

INTRODUCTION: Authors give a detailed background about the biological activities of the extracts obtained by olive leaves, but principally about phenolic fraction. They should improve information about the potential biological activities of the terpenic fraction.

FIG 5. There are some intense peak not identified. Have authors tried identification with different MS data bank? Have they tried by studying literature data. In any case if the result of identification is not gained they have to report them in table as unknowns.

TABLE 4 Please indicate in the legend the name of the MASS BANK used for identification. If more Databases are used a legend with the correct specification is necessary.

CONCLUSIONS

Conclusions are totally focused on the technical approach applied to improve the SFE extraction. Is it  possible to have some input about the interest of have a characterization of terpens in this waste material, for their biological activity with the aim of an industrial use?

Author Response

The paper “Mass Spectrometric Characterization of Terpenes Recovered from Olive Leaves Using Adsorbent-Assisted Supercritical CO2 Fractionation” reports an experimental study focused to the extraction with SFE (Adsorbent –assisted) COfollowed by a GC-QTof MS analysis with characterization based on Mass Spectra Databases.

In my opinion the paper can be accepted for publication after minor revision.

We want to thank the reviewer for considering that our work can be accepted after minor revision.

TITLE: In my opinion, the authors have to revise the title. By the present title seems that Mass Spectrometry characterization is central in the experimental work, but it is not (technical development of Adsorbent-Assisted Supercritical CO2 extraction is the central goal of the paper). Fractionation has to be changed in extraction, because it seems authors have fractionated the extract by density stepping or other procedures, while the object of the GC-MS analysis is a single extract, as showed in figure 5.

According to reviewer suggestion, we have slightly modified the title to: “Extraction and Mass Spectrometric Characterization of Terpenes Recovered from Olive Leaves Using A New Adsorbent-Assisted Supercritical CO2 Process” in order to emphasize the two key aspects of the manuscript: extraction and chemical characterization. In our opinion, both parts are essential since the central goal was to obtain and characterize different fractions from olive leaves with a tailored composition in terpenes.

INTRODUCTION: Authors give a detailed background about the biological activities of the extracts obtained by olive leaves, but principally about phenolic fraction. They should improve information about the potential biological activities of the terpenic fraction.

In this particular aspect, we want to emphasize that in total 31 references can be found in the manuscript, 16 publications included in the Introduction related to biological activities associated to the terpene fraction of olive leaves (References 6-9, 17-28) and 15 references cited in the Results and Discussion section, including biological activities of certain terpenes (References: 42-56) that were identified in the extract. Therefore, we consider that the amount of information is sufficiently focused on terpenoids, which are the target compounds of this manuscript.

FIG 5. There are some intense peak not identified. Have authors tried identification with different MS data bank? Have they tried by studying literature data. In any case if the result of identification is not gained they have to report them in table as unknowns.

We want to thank the reviewer for his/her comment. In fact, all the intense peaks have been identified; we did not include this information in the previous version since our focus was to identify terpenoids in the extract. In order to include the information, we have modified Figure 5 with the identification of the rest of the peaks and include this information in the legend. Therefore, Table 4 has been left unchanged since it is only referring to terpenoids and gathers all the information about these compounds in olive leaves.   

TABLE 4 Please indicate in the legend the name of the MASS BANK used for identification. If more Databases are used a legend with the correct specification is necessary.

Done.

CONCLUSIONS

Conclusions are totally focused on the technical approach applied to improve the SFE extraction. Is it possible to have some input about the interest of have a characterization of terpens in this waste material, for their biological activity with the aim of an industrial use?

As the reviewer mentions, considering terpenoids as target compounds identified in the manuscript, we have worked also in the biological activities, in this case related to the neuroprotective effect of the extracts enriched in target terpenes; this information is reflected in a second manuscript that has been sent for publication into Foods as a second part of the present manuscript (Manuscript ID: foods-1247012; Title: Neuroprotective Effect of Terpenoids Recovered from Olive Oil By-Products). Considering the huge amount of information, we decided to divide the manuscript in two parts.

Moreover, we have added a comment in the conclusions, as suggested: “Future work evaluating the neuroprotective activity of the different olive leaves’ extracts is being carried out in our laboratory.”

Reviewer 4 Report

The manuscript is overall well written and interesting. The methodology seems sound and the conclusions are supported by the reported results. The aspect of novelty was clearly outlined in the Introduction section. The subject matter is relevant to the scope of Foods.

A general remark: the Authors should carefully proofread the manuscript, as it contains some language errors.

Detailed remarks:

- lines 18-19: it's not that the adsorbents were used to assess their selectivity, but rather their selectivity that was assessed/evaluated;

-line 22: QTOF

-lines 59-60: I'm assuming these are varieties of olives, not leaves?

- line 221: Agilent 7200 quadrupole time-of-flight (Q-TOF) detector/analyser?

- line 225: which version of the NIST database?

- subsection 2.6: please provide additional details regarding the multivariate statistical analysis (normalization, what sort of clustering was used, the type of linkage and distance calculation method, etc.);

- table 2: the '2' in CH2OH should be indexed;

- fig.5, while legible when magnified, might be difficult to read in print. Perhaps the authors would consider placing it on a horizontally aligned page, similar to some of the tables?

Author Response

The manuscript is overall well written and interesting. The methodology seems sound and the conclusions are supported by the reported results. The aspect of novelty was clearly outlined in the Introduction section. The subject matter is relevant to the scope of Foods.

We want to thanks reviewer 2 for his/her positive comments on our manuscript.

A general remark: the Authors should carefully proofread the manuscript, as it contains some language errors.

A detailed proofreading of the manuscript has been done, as suggested.

Detailed remarks:

- lines 18-19: it's not that the adsorbents were used to assess their selectivity, but rather their selectivity that was assessed/evaluated;

Text has been modified accordingly.

-line 22: QTOF

Done.

-lines 59-60: I'm assuming these are varieties of olives, not leaves?

Reviewer is absolutely right. This aspect has been clarified.

- line 221: Agilent 7200 quadrupole time-of-flight (Q-TOF) detector/analyser?

Analyzer was added.

- line 225: which version of the NIST database?

NIST 20 Mass Spectral database was used for MS search.

- subsection 2.6: please provide additional details regarding the multivariate statistical analysis (normalization, what sort of clustering was used, the type of linkage and distance calculation method, etc.);

For multivariate data analysis, a compound-abundance table, including samples in columns, was submitted to cluster analysis and heatmapping using freely available web server Heatmapper (www.heatmapper.ca). Data matrix was previously scaled using an auto-scaling approach; that is, data was mean-centered and divided by the standard deviation of each variable. A hierarchical clustering was applied using a complete linkage clustering method with Pearson distance measurement.

- table 2: the '2' in CH2OH should be indexed;

Done.

- fig.5, while legible when magnified, might be difficult to read in print. Perhaps the authors would consider placing it on a horizontally aligned page, similar to some of the tables?

Done.

Round 2

Reviewer 1 Report

I believe the manuscript has been sufficiently improved